# D2P2-SGD: Dynamically Differentially Private Projected Stochastic Gradient Descent

## Abstract

Stochastic optimization is a key enabler in modern machine learning, producing effective models for various tasks. However, several researchers have shown that model parameters and gradient information are susceptible to privacy leakage. Although, Differentially Private SGD (DPSGD) addresses privacy concerns, its static noise mechanism impacts the error bounds for model performance. Additionally, with the exponential increase in model parameters, efficient learning of these models using stochastic optimizers has become more challenging. To address these concerns, we introduce the Dynamically Differentially Private Projected Stochastic Gradient Descent (D2P2-SGD) optimizer. In D2P2-SGD, we combine two important ideas: (i) dynamic differential privacy (DDP) with automatic gradient clipping and (ii) random projection with SGD, allowing dynamic adjustment of the tradeoff between utility and privacy of the model. It demonstrates provably tighter error bounds compared to DPSGD across different behavior (i.e. convex and non-convex) of the objective function. The theoretical analysis further suggests that DDP leads to better utility at the cost of privacy, while random projection enables more efficient model learning. Extensive experiments across diverse datasets show that D2P2-SGD significantly enhances accuracy while maintaining privacy. Our code is available here.

## 1 Introduction

Deep learning models Kasneci et al. (2023); Chang et al. (2023); Thirunavukarasu et al. (2023); Zhao et al. (2023); Menghani (2023), enabled by stochastic optimization techniques have achieved remarkable success in many fundamental machine learning tasks. Though these models empirically show incredibly appealing capabilities, there are critical concerns regarding the privacy of these models. In numerous applications, such as healthcare Chen et al. (2021) and finance Goodell et al. (2021), training datasets often contain highly sensitive information that must remain confidential. However, due to the wide use of deep learning models, their rich representations can possibly disclose private information under privacy attacks, as demonstrated in the prior works Zhu et al. (2019); Zhao et al. (2020); Wang et al. (2022).

To mitigate these privacy concerns, *differential privacy* (DP) Dwork (2006; 2008); Li et al. (2011) was introduced, and it has gained considerable attention Ji et al. (2014); Blanco-Justicia et al. (2022) to provide principled and rigorous privacy guarantees. Intuitively speaking, DP is a mechanism to ensure that all data samples have no significant impact on the ultimate trained model. Differentially private SGD (DPSGD) Song et al. (2013); Jagielski et al. (2020); Abadi et al. (2016); Bassily et al. (2014) is one of the most acknowledged methods to solve the private empirical risk minimization (ERM) problems. Specifically, it perturbs each gradient update with a static random noise vector (with the same dimension as that of the gradient) sampled from a distribution. Using the perturbed gradient updates, we can compute the tradeoff between the utility and privacy of the model. This tradeoff can be adjusted via a noise mechanism, typically sampled from a *static* distribution with properly chosen yet fixed variance. While this approach is technically simple and provably effective, the noise variance significantly impacts the ultimate error bound.

A challenging issue that remains in various private machine learning tasks is how to achieve a desirable tradeoff between privacy and utility, which becomes particularly important in large-scale deep learning models Li et al. (2021a); Flemings et al. (2024); Mattern et al. (2022).

Recent work Du et al. (2021) proposed a dynamic DP mechanism to adjust the tradeoff on the fly, reducing the model performance loss gap but at the cost of increased privacy loss. Further, the dimension of the noise vector is typically the same as that of the gradient, which could be extremely large to cause intensive computational complexity. This issue motivates us to seek out an approach that assists in reducing the complexity while maintaining privacy. Inspired by the work Blocki et al. (2012), we pay attention to model compression techniques.

*Model compression* Buciluă et al. (2006); Choudhary et al. (2020) has been utilized to reduce the computational complexity, including quantization Zhou et al. (2018); Chmiel et al. (2020), regularization Moradi et al. (2020); Orvieto et al. (2023); Tao et al. (2015), and projection Gu et al. (2023); Tsfadia (2024). Though these methods aim to reduce computational complexity, their technical details can differ significantly depending on the specific focus. For example, in Gu et al. (2023), the authors used projection to identify the dominating gradient subspace, which assisted in improving model accuracy, without changing the dimension of model parameters. This is analogous to L1-norm regularization Xu et al. (2008), which enforces some model parameters to be exactly 0. Albeit model compression enables enticing performance, it comes at the expense of possible accuracy reduction and sophisticated compression techniques, necessitating an effective optimizer that can balance the dynamic trade-off between privacy and utility. Thus, this naturally leads to a synthesis among complexity, privacy and utility and the following question:

*Can we design an efficient differentially private optimizer to allow small model performance reduction gap and maintaining privacy?*

**Contributions.** In this work, we answer this question affirmatively. Specifically, we propose a novel stochastic optimizer termed Dynamically Differentially Private Projected Stochastic Gradient Descent (D2P2-SGD), which, for the first time, integrates the dynamic DP mechanism with automatic gradient clipping and random projection for optimization. The dynamic DP mechanism involves an isotropic Gaussian distribution with a properly chosen *time-varying* variance that decreases along with the iterations, reducing noise effects as privacy loss increases. To further warrant differential privacy and bound the influence of each individual example on the stochastic gradient, we resort to a recently developed automatic gradient clipping mechanism Bu et al. (2024). This is different from traditional clipping method Abadi et al. (2016) that there is an upper bound imposed for gradients. Additionally, the stochastic gradient is projected into a lower-dimensional space, reducing the dimension of additive noise and mitigating the increase in privacy loss. Concretely, the main contributions are as follows:

1. We propose a novel DP optimizer, D2P2-SGD, by defining a dynamic DP mechanism with time-varying noise variance and using random projection to reduce the model performance loss gap and the dimension of noise vectors added to stochastic gradients. The per-sample gradient normalization is utilized as the automatic gradient clipping mechanism to control the influence of each individual example on the stochastic gradient.
2. Theoretically, we show that with D2P2-SGD, the error rates for generally convex and non-convex functions are *tighter* than those for regular DPSGD. The results for the dynamic DP mechanism can immediately degenerate to those for the static scenario, revealing the consolidation among complexity, utility, and privacy.
3. We validate the proposed D2P2-SGD on a wide spectrum of datasets, demonstrating that the model accuracy can substantially improve compared to the state-of-the-art.

It is noteworthy that our approach in this paper prioritizes theoretical exploration over scalability to larger datasets and models, such as transformers and large language models Kasneci et al. (2023); Chang et al. (2023). While some work has been done in this area Anil et al. (2021); Yu et al. (2021a), developing a differentially private optimizer for these models remains a significant challenge and is left for future work.

## 2 RELATED WORKS

Beyond the aforementioned works, additional related works have been done to develop more efficient DP optimizers. The authors in Koloskova et al. (2023b) designed a novel optimizer called anti-correlated perturbed gradient descent (Anti-PGD) and analyzed its convergence rates for diverse functions. Though they claimed the bounds are tighter and used the results to develop new and

Table 1: Comparison among different methods.

| Method | N. | C. | Rate |
|---|:---:|:---:|:---:|
| DPSGD Bassily et al. (2014) | S | N | $\mathcal{O}(\frac{1}{\sqrt{K}})$ |
| PDP-SGD Zhou et al. (2020) | S | Y | $\mathcal{O}(\frac{1}{\sqrt{K}})$ |
| DPKD Mireshghallah et al. (2022) | S | N | N/A |
| Anti-PGD Koloskova et al. (2023b) | S | N | $\mathcal{O}(\frac{1}{\sqrt{K}})$ |
| Dynamic DPSGD Du et al. (2021) | D | N | $\mathcal{O}(\frac{1}{\sqrt{K}})$ |
| PrivSGD Kasiviswanathan (2021) | S | Y | N/A[1] |
| RQP-SGD Feng & Venkitasubramaniam (2024) | S | Y | $\mathcal{O}(\frac{1}{\sqrt{K}})$ |
| D2P2-SGD (Ours) | D | Y | $\mathcal{O}(\frac{1}{\sqrt{K}} + \frac{\ln K}{K^{1.5}})$ |

S: static; D: dynamic; N: no; Y: yes; [1] N/A: the convergence rate in this work is only for convex; however, the rate herein is for non-convex functions. N.: Noise. C.: Compression.

effective matrix factorizations for DP optimization, no relevant theoretical results have been found for the trade-off between utility and privacy. A recent work for the first time (and possibly being only one to our best of knowledge) proposed a dynamic DP mechanism Du et al. (2021) to vary the noise variance along with the optimization. In this work, the authors scrutinized how to dynamically adjust the gradient clipping thresholds and noise power for alleviating the performance loss gap given a total privacy budget constraint. However, they overlooked the dependence of noise vector on the dimension of gradient, which can be extremely large to worsen the performance loss gap. Moreover, regardless of delivering meaningful insights for the trade-off between utility and privacy, no formal results were reported on the privacy guarantee. A projected variant of DPSGD Zhou et al. (2020) provably showed the tighter error bounds based on the reduced dependence on the originally large dimension by identifying the gradient sub-eigenspace. Although the proposed scheme is mathematically simple, calculating eigenvectors from structure tensors could become problematic if they are ill-conditioned. Furthermore, a small public dataset was entailed to estimate structure tensors, which may not be satisfied if their method was applied to a distributed setting. A work close to ours is PrivSGD in Kasiviswanathan (2021), which used random projection on the gradient vector to reduce the dependence on the large dimension arising from the noise addition. Nevertheless, in this work, an extra optimization was required to lift the lower-dimensional gradient back to the original one, which inevitably caused more computational overhead. In addition to this, the noise power in the DP mechanism still remained static. The authors in Yu et al. (2021b) proposed Gradient Embedding Perturbation (GEP) to first project gradients into a non-sensitive anchor subspace and then perturb the low-dimensional embedding and the residual gradient separately according to the privacy budget. Though this work has implemented low-dimensional embedding to reduce the perturbation variance, as we do in this work, there is no control mechanism on the gradient that may diverge. Also, the utility analysis only applies to convex case. A more recent work Feng & Venkitasubramaniam (2024) employed randomized gradient quantization to compress the dimension of the gradient vector for reducing the computational complexity, facing the same issues including large dimension and static noise power. Different from these existing works, our work aims to reduce the performance loss gap by attenuating the negative noise impact without significantly sacrificing the privacy. Please refer to Table 1 for a summarized comparison among different methods.

## 3 PROBLEM FORMULATION AND PRELIMINARIES

Given a private dataset $\mathcal{D} = \{s_1, s_2, ..., s_n\}$ sampled in an i.i.d. manner from a distribution $\mathcal{P}$ such that we want to solve the empirical risk minimization (ERM) problem subject to differential privacy:

$$\min_{\mathbf{x}} f(\mathbf{x}) = \frac{1}{n} \sum_{i=1}^{n} f(\mathbf{x}, s_i), \tag{1}$$

where $\mathbf{x} \in \mathbb{R}^d$ and $f(\cdot, \cdot)$ is the loss for a single sample. We aim to optimize Eq. 1 with a gradient-based algorithm in a differentially private manner. We denote by $\mathbf{x}_k$ the model parameters' iterate and $\mathbf{g}_k$ the mini-batch gradient at each time step $k$. Throughout the analysis, we assume that $\mathbf{g}_k$

is the unbiased estimate of $\nabla f(\mathbf{x}_k)$. In this context, we resort to *gradient clipping mechanism* to constrain the magnitude of the stochastic gradient $\mathbf{g}_k$. To provide the guarantee of differential privacy, it requires bounding the influence of each individual example on $\mathbf{g}_k$. A fairly popular clipping operation Abadi et al. (2016) applied to vector $\mathbf{v} \in \mathbb{R}^d$ is as: $\text{clip}(\mathbf{v}, G) = \min\{1, \frac{G}{\|\mathbf{v}\|}\} \cdot \mathbf{v}$, where $G > 0$, $\|\cdot\|$ is the $l_2$ norm. However, when applying this to the stochastic gradient, such an operation will inevitably result in "lazy region" issue, particularly if $\|\mathbf{v}\| > G$. This means the parameters will not be updated even if the true gradients are non-zero. Therefore, to mitigate this issue, we leverage a recently developed *per-sample gradient normalization* Bu et al. (2024) as an automatic clipping described as: $\text{clip}(\mathbf{v}, G, \gamma) = \frac{G}{\|\mathbf{v}\| + \gamma}$, where $\gamma$ is a positive stability constant, which has been recommended in their work. Additionally, the authors even showed that any constant choice $G$ is equivalent to choosing $G = 1$. In this work, we reveal that for any $\gamma > 0$, the gradient norm will converge to a neighborhood at the same asymptotic rate such that common deep learning optimizers are insensitive to the choice of $\gamma$. To characterize the analysis for the propose scheme, we introduce the necessary background and preliminary knowledge in the sequel, starting with the standard definition of differential privacy.

**Definition 1.** *($\varepsilon$-Differential Privacy Dwork (2006)) A randomized algorithm $\mathcal{M}$ is $\varepsilon$-differentially private if for any pair of datasets $\mathcal{D}, \mathcal{D}'$ differ in exactly one data point and for all events $\mathcal{Y} \subseteq Range(\mathcal{M})$ in the output range of $\mathcal{M}$, we have $Pr\{\mathcal{M}(\mathcal{D} \in \mathcal{Y})\} \leq \exp(\varepsilon)Pr\{\mathcal{M}(\mathcal{D}' \in \mathcal{Y})\}$, where the probability is taken over the randomness of $\mathcal{M}$.*

$Range(\mathcal{M})$ refers to the set of all possible outcomes of $\mathcal{M}$. Technically speaking, the set $\mathcal{Y}$ in Definition 1 must be measurable. This definition implies that the probability of observing a specific output on any two neighboring datasets can differ by at most a multiplicative factor of $\exp(\varepsilon)$. Intuitively, a sufficiently small $\varepsilon$ value suggests that either including or excluding a single data point from the dataset does not likely affect the output. Hence, an adversary only accessing the output of $\mathcal{M}$ is difficult to infer whether any data point is present in the dataset. The parameter $\varepsilon$ is called *privacy budget* and its practical selection varies significantly, depending on different scenarios Ponomareva et al. (2023). However, a relaxation of $\varepsilon$-DP in Definition 1 has been used more commonly instead, which is primarily due to attaining better utility and easier privacy accounting for composing multiple DP mechanisms. Consequently, an *Approximate* DP mechanism is defined as follows.

**Definition 2.** *(($\varepsilon, \delta$)-differential privacy Dwork (2006)) A randomized algorithm $\mathcal{M}$ is $(\varepsilon, \delta)$-differentially private if for any two neighboring datasets $\mathcal{D}, \mathcal{D}'$ and for all events $\mathcal{Y} \subseteq Range(\mathcal{M})$ in the output range of $\mathcal{M}$, we have $Pr\{\mathcal{M}(\mathcal{D} \in \mathcal{Y})\} \leq \exp(\varepsilon)Pr\{\mathcal{M}(\mathcal{D}' \in \mathcal{Y})\} + \delta$, where the probability is taken over the randomness of $\mathcal{M}$.*

Here, it can be observed that $\delta$ controls the strength of the relaxation, compared to Definition 1, with smaller values leading to stronger privacy guarantees. A generally recommended $\delta$ value in the literature is to choose $\delta \ll \frac{1}{n}$ Ponomareva et al. (2023). In our analysis, we will establish the privacy guarantee for the proposed algorithm presented in the next section. In the sequel, we present preliminaries on random projection and define formally the projection matrix for our algorithm.

Random projection (RP) Achlioptas (2001) is an effectively fundamental tool that has been used in numerous applications to analyze datasets and then characterize their major features. It projects data points to random directions that are independent on the dataset, which renders simpler and computationally faster trend than classical methods such as singular value decomposition (SVD). RP is based upon the Johnson-Lindenstrauss lemma Larsen & Nelson (2017), described as follows.

**Definition 3.** *(Johnson-Lindenstrauss Lemma Larsen & Nelson (2017)) For any $0 < \zeta < 1$, a set $\mathcal{S}$ of $m$ points in $\mathbb{R}^d$, and an integer $p > 8(\ln m)/\zeta^2$, there exists a linear map $h : \mathbb{R}^d \to \mathbb{R}^p$, such that*

$$(1 - \zeta)\|u - v\|^2 \leq \|h(u) - h(v)\|^2 \leq (1 + \zeta)\|u - v\|^2, \tag{2}$$

*for all $u, v \in \mathcal{S}$.*

Johnson-Lindenstrauss lemma states that a set of points in a high-dimensional space can be projected into a lower dimension subspace such that their relative distances are nearly preserved. Inspired by this, we adapt random projection techniques to model parameters or gradients Kasiviswanathan (2021) by projecting the counterpart from a high-dimensional space to its corresponding subspace, which facilitates the efficient updates. Note also that the lower dimension subspace is selected randomly based on some distribution. According to Definition 3, we observe that the core to the

Johnson-Lindenstrauss Lemma is the linear map $h$, which can be obtained through the following definition.

**Definition 4.** *Let $A$ be a random matrix of order $d \times p$, i.e., $A_{ij} \sim \mathcal{N}(0, 1)$ and $o$ be any fixed vector in $\mathbb{R}^d$. Define $r = \frac{1}{\sqrt{p}} A^\top o$. Thus, $r \in \mathbb{R}^p$ and $r_i = \frac{1}{\sqrt{p}} \sum_j A_{ij} o_j$.*

We notice that each element of $A$ is sampled from the same normal distribution $\mathcal{N}(0, 1)$, while we use a slightly different variance, $\sigma_A^2$ instead of 1, in our theoretical analysis.

## 4 Algorithm and Main Results

We present the main algorithm framework for the proposed D2P2-SGD and defer the variants to the supplementary materials.

### 4.1 Algorithmic Frameworks

D2P2-SGD is shown in Algorithm 1. Line 4 states the key gradient clipping operation to control the influence of the gradient magnitude. Compared to the clipping mechanism applied in Abadi et al. (2016), we do not need to tune the clipping threshold. In Line 5, a mini-batch stochastic gradient is calculated after the per-sample gradient clipping. In Line 6, the stochastic gradient $\mathbf{g}_k$ is projected to the lower-dimensional space $\mathbb{R}^p$ by using $A_k$, which is followed by adding the noise sampled from a Gaussian distribution with time-varying distribution $\sigma_{\epsilon,k}^2 \mathbb{I}_p$, where $\sigma_{\epsilon,k} = \frac{\sigma_\epsilon}{\sqrt{k}}$. With this, $\epsilon_k$ is independent of a high dimension $d$, but dependent on a lower dimension $p \ll d$, which fundamentally reduces the noise. The fact that we resort to the decay of $\frac{1}{\sqrt{k}}$ for the variance is motivated by the same setup to the learning rate in stochastic optimizers Bottou et al. (2018), which manipulates the tradeoff between the convergence speed and optimality. Analogously, $\sigma_{\epsilon,k}^2$ controls the impact of the noise mechanism on the tradeoff between the privacy and utility in different phases of the optimization. Since the update for $\mathbf{x}_k$ is operated in the original dimension $\mathbb{R}^d$, we multiply the projected stochastic gradient by $A_k$ to project it back to the original one. It is apparent that such an implementation will cause projection errors that will impact the error rate, (which will be observed in the theoretical analysis). However, similar to Wang et al. (2019), D2P2-SGD implies more efficient model learning as it has now focused primarily on the subspace in $\mathbb{R}^d$ instead of the whole space. Note that the temporal evolution of $A_k$ is due to its elements sampled from a constant distribution every iteration. We claim that D2P2-SGD represents a unified framework over existing methods. When $p = 1$ and $A_1 = A_2 = ... = A_K = I$, D2P2-SGD degenerates to dynamic DPSGD (D2P-SGD) Du et al. (2021), though the original approach has another gradient clipping mechanism to prevent dynamic DPSGD from diverging and different formula for $\sigma_{\epsilon,k}^2$. Similarly, if we set fixed variance for $\epsilon_k$, D2P2-SGD becomes DPSGD, without any random projection. On the contrary, PrivSGD Kasiviswanathan (2021) can also be obtained if D2P2-SGD has the fixed variance, but with the random projection. However, compared to PrivSGD, which involves an extra optimization to convert from the low-dimensional to high-dimensional spaces, our scheme simply uses $A_k$ to replace the optimization, which significantly attenuates the practical implementation complexity. We also use DP2-SGD (differentially private projected SGD) to represent this case. Please see all the methods in the supplementary materials.

---

**Algorithm 1** D2P2-SGD

1: **Input:** Model initialization $\mathbf{x}_1$, step size $\alpha$, the number of epochs $K$, lower dimension $p$, random matrices $A_1, A_2, ..., A_K$, size of mini-batch $B$, training set $\mathcal{D}$, noise sequence $\sigma_{\epsilon,1}^2, \sigma_{\epsilon,2}^2, ..., \sigma_{\epsilon,K}^2, \gamma$
2: **for** each $k$ in 1 to $K$ **do**
3:      Split the dataset $\mathcal{D}$ into multiple mini-batches with size $B$ and randomly sample one $\mathcal{B}$
4:      Clip the per-sample gradient $\hat{\mathbf{g}}_k^s = \nabla f(\mathbf{x}_k; s) / (\|\nabla f(\mathbf{x}_k; s)\| + \gamma)$, $s \in \mathcal{B}$
5:      Calculate the mini-batch stochastic gradient $\mathbf{g}_k = \frac{1}{B} \sum_{s \sim \mathcal{B}} \hat{\mathbf{g}}_k^s$
6:      Project noisy gradient using $A_k^\top$: $\tilde{\mathbf{g}}_k = A_k(\frac{1}{\sqrt{p}} A_k^\top \mathbf{g}_k + \epsilon_k)$, where $\epsilon_k \sim \mathcal{N}(0, \sigma_{\epsilon,k}^2 \mathbb{I}_p)$
7:      Update parameter using projected noisy gradient: $\mathbf{x}_{k+1} = \mathbf{x}_k - \alpha \tilde{\mathbf{g}}_k$
8: **end for**
9: **Output:** $\mathbf{x}_K$

---

## 4.2 MAIN RESULTS

We next show the convergence behavior for our proposed D2P2-SGD, with generally convex and non-convex objective functions. All proof is deferred to the appendix. Before presenting the results, we impose some necessary assumptions.

**Assumption 1.** *(a): $f(\mathbf{x})$ is smooth with modulus $L$ for all $\mathbf{x} \in \mathbb{R}^d$ and coercive; (b) throughout the analysis, the minimum value of the objective $f$ exists and is bounded below, i.e., $f^* := f(\mathbf{x}^*), \mathbf{x}^* = \min_{\mathbf{x} \in \mathbb{R}^d} f(\mathbf{x})$ and $f^* > -\infty$.*

Assumption 1 (a) is generic in many previous works Wang et al. (2019); Du et al. (2021); Kasiviswanathan (2021) to imply that the variations of gradients along with optimization is bounded above by $L$. Many models even including deep neural networks can at least be approximately smooth for the corresponding losses.

**Assumption 2.** *(a) The variance of stochastic gradient $\nabla f(\mathbf{x}, s)$ is bounded above by a constant $\sigma > 0$, i.e., $\mathbb{E}[\|\nabla f(\mathbf{x}, s) - \nabla f(\mathbf{x})\|] \leq \sigma^2, \forall s \in \mathcal{D}$; (b) for any $\mathbf{x} \in \mathbb{R}^d$ and any sample $s \in \mathcal{D}$, $\|\nabla f(\mathbf{x}, \mathbf{s})\| \leq G$.*

Assumption 2 (a) is popular when analyzing the convergence behavior of SGD type of algorithms due to the mini-batch sampling. As in Wang et al. (2019); Du et al. (2021), the bounded gradient in Assumption 2 (b) is *only* needed for of utility guarantee of generally convex objectives, which has typically been regarded as a strong assumption. However, it was also used in recent works Zhou et al. (2020); Zhang et al. (2023). The author in Kasiviswanathan (2021) used an extra bounded second moment assumption for gradients besides the bounded variance assumption, though it is weaker than the bounded gradient assumption. Additionally, to measure the convergence, we adopt function difference $f(\mathbf{x}) - f^* \leq \xi$ for generally convex functions and the norm of gradient $\|\nabla f(\mathbf{x})\| \leq \xi$ for non-convex functions, where $\xi > 0$ can be an arbitrarily small constant. In the sequel, we start the main results with the privacy guarantee.

**Theorem 1.** *(Privacy) Let Assumption 2 (b) hold. There exist constants $C_1, C_2 > 0$ such that for any $\varepsilon \leq \frac{C_1 B^2 K}{n^2}$, D2P2-SGD is $(\varepsilon, \delta)$-differentially private for any $\delta > 0$, if $\sigma_\epsilon^2 \geq \frac{C_2 K^2 \ln(1/\delta)}{n^2 \varepsilon^2}$.*

The proof is essentially adapted from the same proof in Abadi et al. (2016), while differing in some constants. At each iteration, Line 7 in Algorithm 1 post-processes the Gaussian noise mechanism that perturbs the stochastic gradient $\mathbf{g}_k$ by adding noise $\epsilon_k$. Subsequently, the sequence of $\{\frac{1}{\sqrt{p}} A_k^\top \mathbf{g}_k + \epsilon_k\}_{k=1}^K$ is released to have privacy guarantee by following the same privacy proof of Theorem 1 in Abadi et al. (2016). The detailed proof is deferred to the Appendix **??**. However, the significant difference in our work is that the noise variance is time-varying, i.e., $\sigma_{\epsilon,k}^2$. With the explicit form of noise variance we have defined in this work, i.e., $\sigma_{\epsilon,k}^2 = \frac{\sigma_\epsilon^2}{k}$, it is immediately obtained that $\sigma_{\epsilon,1}^2 > \sigma_{\epsilon,2}^2 > ... > \sigma_{\epsilon,K}^2$. In Wang et al. (2019) and Abadi et al. (2016), the static variance has the lower bound with respect to some key constants such as $K$ and $G$. Thus, as long as $\sigma_{\epsilon,K}^2 \geq \frac{C_2 K \ln(1/\delta)}{n^2 \varepsilon^2}$, the privacy guarantee is attained. Equivalently, $\sigma_\epsilon^2 \geq \frac{C_2 K^2 \ln(1/\delta)}{n^2 \varepsilon^2}$ in this context. Another observation from Theorem 1 is that the size of mini-batch $B$ has an impact on $\varepsilon$. When $B$ enlarges, $\varepsilon$ has a larger upper bound such that the model performance improves with the cost of privacy, which will be evidently validated in the result section. Though the authors in Du et al. (2021) for the first time proposed to leverage dynamic DP mechanism to reduce the model performance loss gap, privacy guarantee has been ensured by the dynamic power following an exponential mechanism $\sigma_{\epsilon,k} \propto \mathcal{O}(\rho^{-\frac{k}{K}})$, where $\rho$ is a positive constant. As they still utilized the clipping mechanism from Abadi et al. (2016), they had also to establish the similar exponential mechanism for the clipping threshold, which makes their algorithm framework more complex. While in our work, thanks to the automatic clipping mechanism, there is no such a requirement. We are now ready to state the results for the utility with different functions.

**Theorem 2.** *(Utility for convex functions) Let Assumptions 1 and 2 hold. Suppose that $f$ is a convex function and that $A$ is a random matrix with each element being sampled from a normal distribution $\mathcal{N}(0, \sigma_A^2)$. Also, let the additive noise of DP mechanism have the variance $\sigma_{\epsilon,k}^2$. If the step size $\alpha = \frac{\alpha_0}{\sqrt{K}}$, where $\alpha_0$ is the base learning rate, and $B = \sigma^2/\xi^2$, then for the iterates $\{\mathbf{x}_k\}_{k=1}^K, K \geq 1$*

*generated by D2P2-SGD, the following relationship holds true*

$$\mathbb{E}[f(\bar{\mathbf{x}}_K) - f^*] \leq \frac{\|\mathbf{x}_1 - \mathbf{x}^*\|^2(G+\gamma)}{2\alpha_0\sqrt{K}\sigma_A^2\sqrt{p}} + \frac{\alpha_0 p^{1.5}(G+\gamma)(\ln K + 1)\sigma_\epsilon^2}{2K^{1.5}} + \frac{\alpha_0 pd^2\sigma_A^2(G+\gamma)}{2\sqrt{K}} + cD\xi,$$

(3)

*where $\bar{\mathbf{x}}_K = \frac{1}{K}\sum_{k=1}^K \mathbf{x}_k, c = \max_k\{\frac{G+\gamma}{\|\mathbf{g}_k\|+\gamma}\} > 1, D = \sup_{\mathbf{x}\in\mathbb{R}^d}\|\mathbf{x} - \mathbf{x}^*\|.$*

**Remark 1.** *Theorem 2 suggests that the error bound involves four terms, the initialization error, the error of additive noise due to the DP mechanism, the random projection approximation error, and the clipping bias. The second term implies the tradeoff between utility and privacy. If $\sigma_{\epsilon,k}^2 = \sigma_\epsilon^2$, this term becomes $\frac{\alpha_0 p^{1.5}c(G+\gamma)\sigma_\epsilon^2}{2\sqrt{K}}$, which result in a slower convergence to vanish it. Instead, if $\sigma_{\epsilon,k}^2 = \frac{\sigma_\epsilon^2}{k}$, it can be bounded by $\mathcal{O}(\frac{\ln K}{K})$, which also relaxes the dependence on $\alpha$ to control the magnitude. Though model performance loss gap is reduced, dynamic variance can breach the privacy. To maintain the $(\varepsilon, \delta)$-differential privacy for D2P2-SGD, as implied in Theorem 1, the additive noise should be sampled with a larger $\sigma_\epsilon^2$ to offset the privacy loss, particularly in the early phase during the optimization, compared to DPSGD. The third term is associated with model projection error, while the term $\sigma_A^2 d^2$ due to model compression can cause significant error. One empirical remedy is to leverage a small $\alpha_0$, leading to the slow convergence. The last term to dictate the error bound in Eq. 3 is a bias caused by the gradient clipping. If the initialization error is small, then D is typically reduced, leading to a smaller bias. Such a bias is attained when batch size B satisfies $B = \mathcal{O}(1/\xi^2)$ Zhao et al. (2021). Without this condition, it would become $cD\sigma$ such that the clipping bias is in the order of $\mathcal{O}(\sigma)$, which resembles the result in Koloskova et al. (2023a). Overall, the consolidation among complexity, utility, and privacy in D2P2-SGD is reflected explicitly in Theorem 2. One trivial corollary summarizes the the clipping bias for D2P2-SGD in the following.*

**Corollary 1.** *(Convergence rate for convex functions) With conditions defined in Theorem 2, the following relationship holds true, i.e., $\mathbb{E}[f(\bar{\mathbf{x}}_K) - f^*] \leq \mathcal{O}(\frac{1}{\sqrt{K}} + \frac{\ln K}{K^{1.5}} + cD\xi).$*

The conclusion in Corollary 1 requires the constant learning rate $\alpha$ to have a format $\alpha \propto \mathcal{O}(\frac{1}{\sqrt{K}})$, which is a quite popular choice in finite time convergence. However, due to the gradient clipping, regardless of how small $\alpha$ is, D2P2-SGD convergences to a neighborhood of size $cD\xi$. This complies with findings from Koloskova et al. (2023a); Bu et al. (2024); Xiao et al. (2023); Chen et al. (2020). However, the bias term we have obtained is quantified by a flexibly smaller constant, such that it can be controlled within a small magnitude of constant. In previous works Xiao et al. (2023); Koloskova et al. (2023a), the clipping bias is tightly correlated with $\sigma$, which can be large under some scenarios. Additionally, Compared to DPSGD with the static variance, our error rate is *tighter*, without sacrificing the privacy, by manipulating the time-varying variance.

**Theorem 3.** *(Utility for non-convex functions) Let Assumptions 1 and 2 hold. Suppose that A is a random matrix with each element being sampled from a normal distribution $\mathcal{N}(0, \sigma_A^2)$. Also, let the additive noise of DP mechanism have the variance $\sigma_{\epsilon,k}^2$. If the step size $\alpha = \frac{\alpha_0}{\sqrt{K}}$, where $\alpha_0$ is the base learning rate, and $B = \sigma^2/\xi^2$, then for the iterates $\{\mathbf{x}_k\}_{k=1}^K, K \geq 1$ generated by D2P2-SGD, the following relationship holds true*

$$\min_{k\in[1,K]}\mathbb{E}[\|\nabla f(\mathbf{x}_k)\|] \leq \frac{f(\mathbf{x}_1)}{\sqrt{K}\sigma_A^2\sqrt{p}\alpha_0} + \frac{\alpha_0 Lp^{1.5}\sigma_\epsilon^2(\ln K + 1)}{K^{1.5}} + \frac{L\alpha_0\sqrt{p}d^2\sigma_A^2}{\sqrt{K}} + 2\xi + \gamma. \quad (4)$$

**Remark 2.** *Viewing Theorem 3 for the non-convex functions allows us to make the similar conclusions as in Remark 1. Nevertheless, different from generally convex cases, the first-order stationary point (FOSP) can only be guaranteed through Eq. 4, which is fairly generic in many stochastic optimization problems. The clipping bias in Eq. 4 is slightly different from that in the generally convex case due to an extra $\gamma$. Similarly, a corollary is presented to show the finite time convergence for D2P2-SGD.*

**Corollary 2.** *(Convergence rate for non-convex functions) With conditions defined in Theorem 3, the following relationships hold true, $\min_{k\in[1,K]}\mathbb{E}[\|\nabla f(\mathbf{x}_k)\|] \leq \mathcal{O}(\frac{1}{\sqrt{K}} + \frac{\ln K}{K^{1.5}} + 2\xi + \gamma).$*

Based on Corollary 2, the convergence rate of non-convex functions remains the same as that of generally convex functions, in a finite time manner. However, comparing results from both cases in the asymptotic manner, the error of non-convex functions may relatively be larger by the stability constant $\gamma$, which illustrates that the asymptotic convergence for non-convex functions is more challenging. We summarize the clipping bias for different methods in the following Table 2 for a comparison

(we only compare for non-convex functions as most of existing methods only discussed non-convex objectives), showing that the clipping bias induced by D2P2-SGD is more flexible to control, when batch size $B = \mathcal{O}(1/\xi^2)$. Also, if the computational complexity is defined as the total number of gradient computation, it can be observed that the computational complexity is $KB = \mathcal{O}(1/\xi^4)$, which retains the same complexity as in Ghadimi & Lan (2013). Similarly, the iteration complexity is $K = \mathcal{O}(1/\xi^2)$, while the convergence is a neighborhood of size $\mathcal{O}(\gamma)$.

Table 2: Comparison among different methods.

| Method | Clipping Bias |
|---|---|
| Chen et al. (2020) | Wasserstein distance |
| Koloskova et al. (2023a) | $\sigma$ or $\sigma^2/a$ |
| Xiao et al. (2023) | $15\sigma$ |
| Bu et al. (2024) | $\sigma/r$ |
| D2P2-SGD (Ours) | $2\xi + \gamma$ |

$a > 0$ is the clipping threshold; $r > 0$

## 5 NUMERICAL EXPERIMENTS

We present extensive empirical results to thoroughly validate our proposed approaches with the comparison to baselines. The baseline we use in this study consists of SGD, vanilla DPSGD, D2P-SGD, and DP2-SGD. D2P-SGD is an equivalent alternative of Dynamic DPSGD in Du et al. (2021). DP2-SGD can also be regarded as an equivalence of PrivSGD Kasiviswanathan (2021) since the compression technique they adopted is also random projection, with a static DP. We leverage the Opacus library Yousefpour et al. (2021) and build the framework on top of it. We use a 4-layer Convolutional Neural Network (CNN) Li et al. (2021b) as the model, which has been widely used in developing optimizers Zhou et al. (2020). A more detailed explanation of the architecture is provided in Appendix. Additionally, the datasets for testing our algorithms include FashionMNIST Xiao et al. (2017) and SVHN Sermanet et al. (2012). As we have particularly identified the critical relationship between the privacy loss $\varepsilon$ and other parameters, an ablation study on this is shown to reveal their impact on the performance. Additional results on other larger models and datasets are in Appendix.

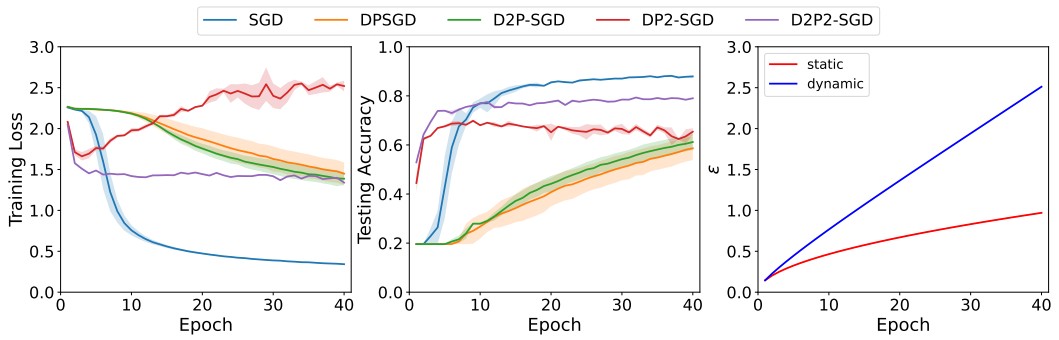

Figure 1: Comparison among different methods for SVHN data: on the right side, the privacy loss is shown for static and dynamic scenarios.

**Comparative Evaluation.** Figure 1 shows the model performance and privacy loss for different methods. We train five different instances of each algorithm with different random seeds. The solid curves correspond to the mean and the shaded region to the minimum and maximum values over the five runs. For the privacy loss, the standard deviation is fairly small. Also, the dynamic mechanism for D2P-SGD and D2P2-SGD is the same. Similarly, DPSGD and DP2-SGD have the same static mechanism. D2P2-SGD significantly improves the model accuracy compared to DPSGD, D2P-SGD, and DP2-SGD. While this comes at the expense of a larger privacy loss, which is expected. This is attributed to the decreasing variance $\sigma_{\epsilon,k}^2/k$ along with iterations. However, the testing accuracy of D2P2-SGD is much closer to SGD, while having a gap due to projection error and clipping bias. Notably, D2P2-SGD spends a lesser number of epochs, reaching a higher accuracy at the early phase, even earlier than SGD, which implies that random projection enables more efficient model learning.

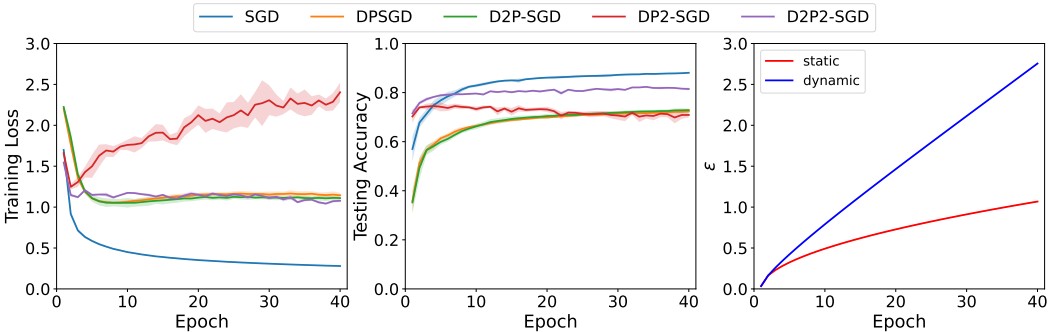

Figure 2: Comparison among different methods for FashionMNIST data: on the right side, the privacy loss is shown for static and dynamic scenarios.

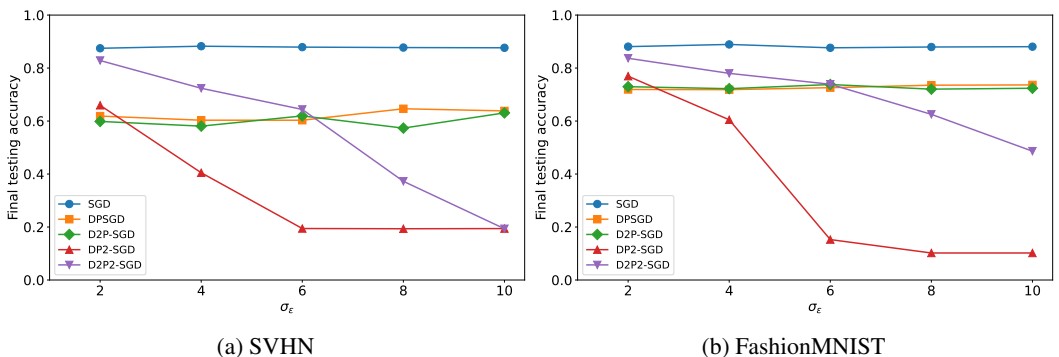

Figure 3: Accuracy vs. standard deviation

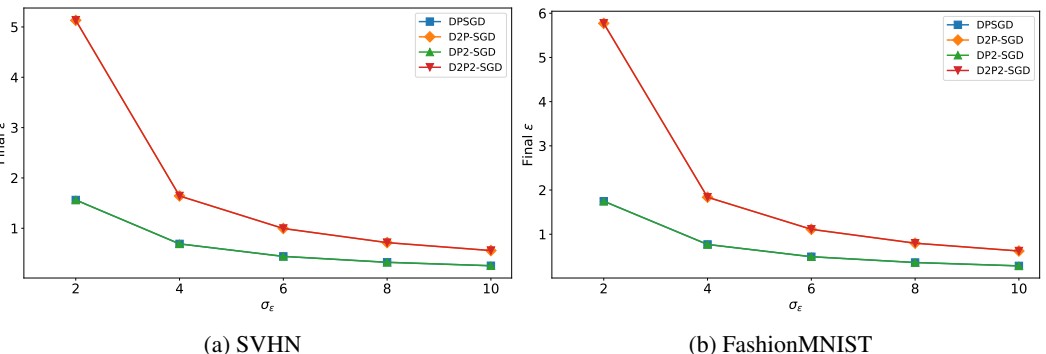

Figure 4: Privacy loss vs. standard deviation

The similar conclusions can be made from Figure 2. Comparing DP2-SGD and D2P2-SGD, we see that with random projection, if the noise variance from the DP mechanism is static, the performance deteriorates accordingly. However, with a dynamic mechanism, it performs robustly throughout training. This validates the conclusion from Theorem 3, where the second term decays faster when the number of iterations increases. However, DP2-SGD remains with the rate of $\mathcal{O}(1/\sqrt{K})$ (If $\alpha$ is not in $\mathcal{O}(1/\sqrt{K})$, this term in D2P2-SGD is in $\mathcal{O}(\ln(K/K))$, while DP2-SGD $\mathcal{O}(1)$). Turning to the privacy loss ($\varepsilon$) in both figures, we can observe that the maximum privacy losses of the dynamic mechanism end up with respectively 2.45 (for SVHN) and 2.75 (for FashionMNIST). Compared to values with the static DP mechanism (1.06 and 0.95, respectively), the privacy loss of D2P2-SGD grows sharply. However, given the bound for $\varepsilon$ in Theorem 1, as long as the constant $C_1$ ($\geq 314$, see supplementary material for more detail) is selected properly, D2P2-SGD still remains $(\varepsilon, \delta)$-differentially private. Thus, our proposed scheme substantially improves the accuracy over baselines while successfully maintaining differential privacy.

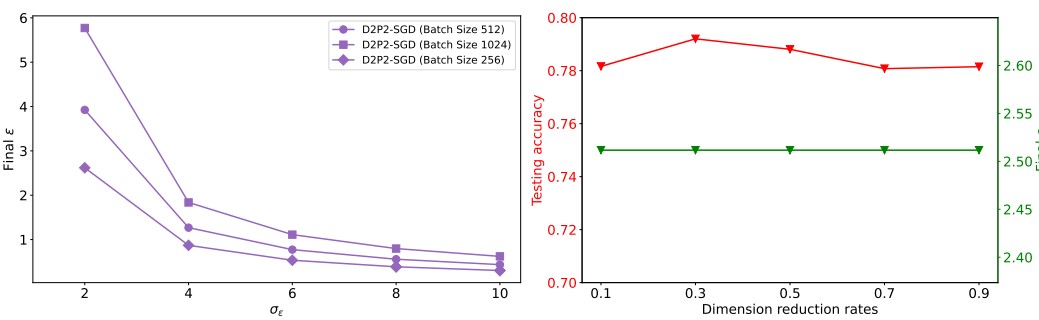

(a) Batch size vs. privacy loss for FashionMNIST  (b) Dimension vs. accuracy/privacy for SVHN

Figure 5: Impacts of parameters

**Impact of $\sigma_\epsilon$.** From Figure 3, it shows that as $\sigma_\epsilon$ value increases, the final model accuracy drops for DP2-SGD and D2P2-SGD, which first validates the coupling between projection error $\sigma_A^2$ and the noise variance $\sigma_{\epsilon,k}^2$ in Theorem 3 and show the trade-off between the utility and privacy. When $\sigma_\epsilon < 6$, which can be treated as a low privacy regime, D2P2-SGD maintains the better performance than DPSGD and D2P-SGD, while underperforming in the high privacy regime after $\sigma_\epsilon > 6$. This intuitively shows us a careful selection of $\sigma_\epsilon$ is required to balance the trade-off. Figure 4 delivers a similar conclusion in terms of privacy loss.

**Impact of $B$.** As suggested from Theorem 1, we can adjust the privacy loss by setting the batch size $B$. In Figure 5a, we can observe that when batch size increases from 256 to 1024, it increases the upper bound for $\varepsilon \leq \frac{C_1 B^2 K}{n^2}$ such that the privacy loss is relatively higher through all $\sigma_\epsilon$ for D2P2-SGD, leading to the better performance. This essentially validates the condition $B = \sigma^2/\xi^2$ from Theorem 2 and Theorem 3, where $\sigma^2$ decreases in the error bounds, leading to the better convergence due to the smaller $\xi$.

**Impact of $p$.** Figure 5b shows the impact of different lower dimensions on the testing accuracy and privacy loss. It immediately suggests that the performance of random projection varies with different $p$ values. The optimal one is a 30% reduction rate, which implies that random projection can assist in model learning efficiency if the $p$ value is chosen properly. Instead, the privacy loss is independent of the dimension change based on Figure 5b. Thus, D2P2-SGD allows for reducing the computational complexity by random projection, while maintaining privacy.

**Limitation.** Though D2P2-SGD has shown good performance compared to the existing baselines, some potential limitations exist, which can also help us close such gaps in future work. First, D2P2-SGD may not work well in the scenarios with *high privacy restrictions*. As we have the decaying noise variance that ensures decent model performance, privacy loss will inevitably be the resulting outcome. One can carefully tune these parameters to obtain acceptable values, but it is still fairly challenging to scale. One way to get rid of this is to develop more effective dynamic DP mechanisms such that the tradeoff between utility and privacy can be handled better. Second, getting an optimal $p$ value for random projection may be difficult as well. Though based on Johnson-Lindenstrauss Lemma, $p$ can be analytically obtained, its practical values for different scenarios have not yet been accessible in a principled manner.

## 6 CONCLUSIONS AND BROADER IMPACTS

This work presents a novel differentially private optimizer termed D2P2-SGD with dynamic DP mechanism, automatic gradient clipping, and model compression, which reveals the synthesis among privacy, utility, and complexity. Specifically, we establish the dynamic privacy guarantee such that a relatively larger variance is required in the early phase of optimization to compensate the privacy loss in the latter phase. Given a pre-defined dynamic variance, D2P2-SGD enables a tighter error bound compared to vanilla DPSGD with static DP mechanism. Empirical results are shown to first validate the theory and then compare with baselines, by using a popular model and benchmark datasets. Compared to vanilla DPSGD, our D2P2-SGD is more robust against larger noise variance, but with a slightly larger privacy loss. However, the accuracy is significantly improved without dependence on the large dimension. Broader vision of this work is to advance the field of differentially private machine learning with potential impact in building privacy-aware deep learning models with highly sensitive information for critical sectors such as healthcare and national security.

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
