# OpenReview forum: "D2P2-SGD: Dynamically Differentially Private Projected Stochastic Gradient Descent"
_ICLR.cc/2025/Conference — ICLR 2025 Conference Withdrawn Submission_

### Official Review · Reviewer_DvkR · 2024-10-31

**Soundness:** 2
**Presentation:** 3
**Contribution:** 1
**Rating:** 3
**Confidence:** 4

**Summary:**

This paper combines several ideas -- automatic clipping, non-uniform privacy budget allocation and JL lemma, on DP-SGD and demonstrates that those improvements can improve the performance.

**Strengths:**

The idea and motivations behind are clearly presented. Proofs look solid and the prior work review is comprehensive. The authors also improved the clipping bias analysis.

**Weaknesses:**

1. Technically, my major concern is that both dimension reduction and automatic clipping are not new ideas. In particular, for DP-SGD with low-dimensional embedded gradients, it has been proved that such technique cannot improve (at least asymptotically) the utility-privacy tradeoff due to the projection bias. As also derived in this paper, D2P2-SGD in Table 1 does not improve the convergence rate but even leads to a worse one.

2. Another mismatch is that the authors argue to use dynamic privacy budget rather than the classic uniform or static selection, while the theorems did not characterize or provide instruction on how to select the budget across the iterations. In the experiments, empirically the authors argue to use a decaying noise but there is no formal analysis to support such selection. A possibly reasonable explanation is maybe the initial stage is more robust to the noise. Also, the authors may want to carefully think about the dynamic protocol: if you want to further adaptively determine the decaying rate based on gradient features, additional privacy budget may be consumed. Anyway, much more work is needed for this dynamic proposal.

3. Only toy examples are presented in this paper. Even from an empirical perspective, the advantage of D2P2-SGD is not very convincing. For example, can the author show comparisons to some SOTA work such as "Unlocking High-Accuracy Differentially Private Image Classification through Scale".

**Questions:**

Please see the above comments.

---

### Official Review · Reviewer_WYGd · 2024-11-03

**Soundness:** 1
**Presentation:** 1
**Contribution:** 2
**Rating:** 3
**Confidence:** 3

**Summary:**

This paper combines two variants of DP-SGD, namely Dynamic DP-SGD (Du et al., 2022) and DP-SGD with auto-clipping (Bu et al., 2024), to propose a privacy-preserving algorithm called D2P2-SGD. The authors claim that the proposed D2P2-SGD achieves a refined privacy-utility trade-off. Experiments are conducted to validate the results.

**Strengths:**

The proposed D2P2-SGD is new to me, even though it is simply a hybrid of two existing methods. Experiments show that the proposed D2P2-SGD achieves higher accuracy on the FashionMNIST and SVHN datasets.

**Weaknesses:**

1. The theory seems confusing and is not self-contained. In fact, to achieve a rate of $O(1/\sqrt{K} + \log^2K/K^{1.5})$, according to Theorem 3 in this paper, it requires that $\sigma_\epsilon^2 = O(\log K)$. However, to achieve differential privacy, in the privacy analysis (Theorem 1), it requires that $\sigma_\epsilon^2 = O(K^2)$. If so, the bound in Theorem 3 will diverge. Additionally, this rate is intuitively confusing. If I understand the four terms in Theorem 3 correctly, the last term should be caused by the auto-clipping techniques, while the remaining terms are from dynamic DP-SGD. Since the dynamic DP-SGD converges at an order of $O(1/\sqrt{K})$, it is unclear why the proposed D2P2-SGD achieves a better utility guarantee. Please clarify this in the rebuttal section.


2. The paper is not well-written, and there are numerous significant typos. Although this is not a reason for my rating, I believe it should be thoroughly polished.

**Questions:**

See the weaknesses part

---

### Official Review · Reviewer_omNo · 2024-11-03

**Soundness:** 3
**Presentation:** 3
**Contribution:** 2
**Rating:** 3
**Confidence:** 4

**Summary:**

The paper proposes a new differentially private gradient-based optimization algorithm. Specifically,
the proposed algorithm introduces two variations to the existing DP-SGD framework. First, it
applies the per-sample gradient normalization to bound the sensitivity of gradient computation.
Second, it projects the normalized gradient to a lower dimensional space using the Gaussian random
projection and the noise is also added in the low-dimensional space. The performance of proposed
algorithm is evaluated on two datasets.

**Strengths:**

- The paper proposes a new differentially private algorithm, called D2P2-SGD, that combines
two techniques: per-sample gradient normalization and random projection.
- The paper analyzes the privacy property and utility of the proposed algorithm and provides
theorems.

**Weaknesses:**

- The proposed algorithm is largely based on existing algorithms and appears to make only
incremental contributions. Specifically, the idea of combining the recently proposed autoclip
techniques with classical random projection approach is interesting but it’s unclear what the
motivation is. Are there any reasons why this combination would outperform existing methods? Besides, recent work [1] empirically demonstrated that the belief that a network with fewer parameters would perform better than a parameter-heavy network is a misconception.

  * [1] De, Soham, Leonard Berrada, Jamie Hayes, Samuel L. Smith, and Borja Balle. “Unlocking High-Accuracy Differentially Private Image Classification through Scale,” arXiv 2022

- The provided proof of privacy property is straightforward due to the use of existing differentially
private techniques. While convergence analysis is provided for both convex and non-convex functions, it neither demonstrates an improvement over prior work nor appears particularly more interesting than those presented in prior work. It seems to follow the same procedure used for analyzing the convergence rate of DP optimizers.
- The empirical results are limited in scope and not well explained. The performance evaluation was done by training a 4-layer CNN on two small datasets FashionMNIST and SVHN. To demonstrate the effectiveness of the proposed approach, more extensive experimental evaluations should be conducted on a variety datasets and architectures.

**Questions:**

- What is the significance of the discussion presented in lines 172 - 174. Isn’t γ a constant used
to ensure the numerical stability of clipping for the gradient with norms close to 0?
- Since the proposed algorithm projects the gradient into a lower dimensional subspace and
adds noise in the projected space, it is natural to include other subspace projection-based
approaches as baselines in the experiments. For example, GEP [1] and PDP-SGD [2]. Why
are they excluded?

  1. Yu, Da, Huishuai Zhang, Wei Chen, and Tie-Yan Liu. “Do Not Let Privacy Overbill Utility: Gradient Embedding Perturbation for Private Learning,” ICLR 2020.
  2. Zhou, Yingxue, Steven Wu, and Arindam Banerjee. “Bypassing the Ambient Dimension: Private SGD with Gradient Subspace Identification,” ICLR 2021.

- I am not sure how to interpret the graph on the right in Figure 1. It seems that the dynamic
mechanism incurs larger privacy loss and ends up with large epsilon values.
- In Figure 3, why does the performance of DP-SGD and D2P-SGD remain almost the same
even when $\sigma_{\epsilon}$ is increased?

---

### Official Review · Reviewer_HUCD · 2024-11-03

**Soundness:** 2
**Presentation:** 3
**Contribution:** 2
**Rating:** 3
**Confidence:** 4

**Summary:**

This paper proposes D2P2-SGD, a differentially private optimizer which combines dynamic DP (i.e., time-varying noise variance) with automatic clipping and random projection. The authors provide a theoretical analysis which demonstrates that the error rates for D2PD-SGD are tighter than for DP-SGD.

**Strengths:**

* D2P2-SGD nicely consolidates several existing methods and I appreciated the “unified framework” angle (line 247).
* The theoretical analysis applies to a general setting, with utility bounds and convergence rates for both convex and non-convex functions.
* The authors provide a very detailed overview of related work and explain how their proposed algorithm differs from previous methods.

**Weaknesses:**

* The paper’s contribution is debatable. While I do think it’s valuable to consolidate existing methods into a unified framework, I wasn’t convinced that D2P2-SGD is greater than the sum of its parts. I also didn’t see much technical novelty in the thoeretical analysis.
* I wasn’t convinced by the empirical evaluation. It looks like the algorithm is only validated on a fairly small number of image datasets. The experimental set-up and the presentation of the results was also confusing for me. For example, I don’t see a “privacy vs utility” plot among the experimental results, which I think is crucial for validating the effectivness of D2P2-SGD compared to existing methods.
* The writing could use polishing; I was a bit distracted by some of the language and I think there are many sentences that could be reworded in order to make them easier to parse.

**Questions:**

* Is there a reason for not including a privacy vs utility plot in the experimental results? For Figures 1 and 2, D2P2-SGD has higher testing accuracy but also higher privacy loss, so I don’t think we can conclude from the figures that D2P2-SGD performs better than other algorithms at a fixed level of privacy. Indeed, from Figures 3 and 4, D2P2-SGD has both a larger privacy loss and lower testing accuracy for $\sigma_{\epsilon} > 6$ and so would appear to be indisputably worse than the baselines at stronger levels of privacy.

* In Figures 1 and 2, the plots on the far right show "epoch vs privacy loss” for static and dynamic mechanisms. Why does the dynamic DP privacy loss increase so much more quickly than the static DP privacy loss?

* Minor detail: the colors in Figure 4 don’t seem consistent with the rest of the figures.

---

### Note · Authors · 2024-11-15

**Comment:**

Thanks, all reviewers for constructive comments!!!! We really appreciate that!!

**Withdrawal Confirmation:**

I have read and agree with the venue's withdrawal policy on behalf of myself and my co-authors.